# Predicting prostate cancer progression: protocol for a retrospective cohort study to identify prognostic factors for prostate cancer outcomes using routine primary care data

Samuel W D Merriel,[1] Margaret T May,[2] Richard M Martin[2]

[1]Centre for Academic Primary Care, University of Bristol, Bristol, UK
[2]Department of Population Health Sciences, University of Bristol, Bristol, UK

**Correspondence to**
Dr Samuel W D Merriel;
sam.merriel@bristol.ac.uk

## ABSTRACT

**Introduction** Prostate cancer is the most common cancer in men in the UK, with nearly 40 000 diagnosed in 2014; and it is the second most common cause of male cancer-related mortality. The clinical conundrum is that most men live with prostate cancer rather than die from it, while existing treatments have significant associated morbidity. Recent studies have shown very low mortality rates (1% after a median of 10-year follow-up) and no treatment-related reductions in mortality, in men with localised prostate cancer. This study will identify prognostic factors associated with prostate cancer progression to help differentiate aggressive from more indolent tumours in men with localised disease at diagnosis, and so inform the decision to adopt conservative (active surveillance) or radical (surgery or radiotherapy) management strategies.

**Methods and analysis** The Clinical Practice Research Datalink (CPRD) contains 57 318 men who were diagnosed with prostate cancer between 1 January 1987 and 31 December 2016. These men will be linked to the Office for National Statistics (ONS) and the National Cancer Registration and Analysis Service registry databases for mortality, TNM stage, Gleason grade and treatment data. Men with a diagnosis date prior to 1 January 1987 and men with lymph node or distant metastases at diagnosis will be excluded. A priori determined prognostic factors potentially associated with prostate cancer mortality, the end point of cancer progression, will be measured at baseline, and the participants followed through to development of cancer progression, death or the end of the follow-up period (31 December 2016). Cox proportional hazards regression will be used to estimate crude and mutually adjusted HRs. Mortality risk will be predicted using flexible parametric survival models that can accurately fit the shape of the hazard function.

**Ethics and dissemination** This study protocol has approval from the Independent Scientific Advisory Committee for the UK Medicines and Healthcare products Regulatory Agency Database Research (protocol 17_041). The findings will be presented in peer-reviewed journals and local CPRD researcher meetings.

---

### Strengths and limitations of this study

► The study cohort is drawn from the Clinical Practice Research Datalink (CPRD), a large, representative UK primary care dataset, with linked cancer registry and Office for National Statistics (ONS) data.
► Predicting cancer progression is a more clinically useful outcome than simply detecting localised disease in patients with a disease that is often indolent and slow-growing.
► Flexible parametric modelling will be used to control for the effects of intermediate variables and prognostic factor combinations will be used.
► Cancer registry data and ONS mortality data are not available for the full study period, and CPRD data may not be complete.

---

## INTRODUCTION

Prostate cancer is the most commonly diagnosed cancer in men, and the second most commonly diagnosed cancer in the UK. A total of 39 741 new cases of prostate cancer were diagnosed in England in 2014, with an age-standardised incidence rate of 177.8 per 100 000 men.[1] In the same year, there were approximately 11 300 prostate cancer deaths in the UK, making prostate cancer the second most common cause of cancer death in men. Nevertheless, survival rates for prostate cancer are relatively high compared with other cancer types. The overall 5-year age-standardised net survival for men with prostate cancer in England was 83.6%, and the predicted 10-year survival for men in England diagnosed with prostate cancer in 2015 is 79.9%.[2] This suggests that many men diagnosed with prostate cancer have indolent disease. A key clinical conundrum relates to distinguishing men with slow-growing tumours that could be managed conservatively with active monitoring from more

BMJ

aggressive, potentially fatal disease that may require more radical intervention.

Prostate cancer can be detected in men in different ways. General practitioners (GPs) need to consider the possibility of prostate cancer in men presenting with lower urinary tract symptoms, erectile dysfunction or visible haematuria. Asymptomatic men may also be found to have raised prostate-specific antigen (PSA) levels and need to be referred for further investigation.[3] However, the use of PSA as a screening and prognostic biomarker remains controversial,[4 5] and GPs[6–9] and patients[10] have mixed views about its utility in informing investigation and treatment decisions for prostate cancer. Other screening methods for predicting prostate cancer severity have been tested, such as the Stockholm-3 (STHLM-3) model,[11] which did not look at risk of progression and relies on genetic biomarkers that are not readily available in primary care at this time.

The Bristol-based *ProtecT* multicentre trial randomised men with clinically localised prostate cancer to either active monitoring, radical surgery (prostatectomy) or radical radiotherapy. After a median of 10-year follow-up there was no difference in prostate cancer mortality. Overall, the 10-year mortality rates were very low (1%), and men randomised to active monitoring were at an increased risk of clinical progression and development of metastatic disease (22.9 per 1000 person years follow-up) compared with the radical treatment arms (8.9 and 9 per 1000 person years, respectively).[12] Men receiving surgery or radiotherapy reported more adverse effects on urinary, sexual and bowel function compared with the active monitoring cohort.[13] This was broadly consistent with other studies of prostate cancer treatment.[14–18] Identifying factors associated with prostate cancer progression may help determine the risk for men having more aggressive prostate cancer, and inform shared decision-making about whether to undergo radical treatments or choose active monitoring.

Cancer progression is well defined in cancer treatment trials, following the widely used RECIST criteria.[19] However, the concept of cancer progression in prognostic studies is much less well defined or consistently applied. The *ProtecT* trial defined prostate cancer progression as the occurrence of any of the following events: evidence of metastasis, development of T3/T4 disease, commencing long-term androgen therapy, ureteric obstruction, rectal fistula and new need for catheter.[12] Several prognostic factors have been identified that may be associated with prostate cancer mortality, the endpoint of prostate cancer progression. These include demographic,[20] genetic,[21] physiological,[22 23] comorbidity,[24–27] lifestyle,[28–33] biochemical[34 35] and medication[36 37] factors. The strength of evidence for these prognostics factors varies and for many others it is conflicting.[38–44]

Primary care medical records contain a wealth of information on a patient's medical history, medications, family history and investigation results.[45] The Clinical Practice Research Datalink (CPRD)[46] is a large UK primary care research database representative of the general population, with links to many other relevant healthcare data registries and Office for National Statistics (ONS) data. This information is already used for many risk prediction tools in primary care settings to predict outcomes and inform treatment decisions. Examples include QCancer,[47] which predicts a patient's absolute risk of future cancer diagnosis. To date there are no risk prediction tools for cancer progression used in clinical practice.

This study aims to establish which risk factors are associated with prostate cancer progression using primary care medical records data. These findings, in combination with metabolomic and genomic data, will inform the development of a clinical risk prediction model for the progression of prostate cancer following diagnosis.

## METHODS AND ANALYSIS

Within the CPRD dataset, at least 57 318 men had a diagnosis of prostate cancer made between 1 January 1987 and 31 December 2016, of whom 22 080 have a recorded date of death. These men will form the basis of the study cohort. Additional mortality, staging (TNM and Gleason grade) and treatment data will be obtained by using each man's National Health Service (NHS) number to link them to the ONS (available from 1 January 1998) and National Cancer Registration and Analysis Service (available from 1 January 1990) databases. The index date will be the date the diagnosis of prostate cancer was first entered into the primary care medical record. From this date, the men will be followed until the date of their death, the development of prostate cancer progression or the end of the cohort period, whichever is later. Men with a diagnosis date prior to 1 January 1987 and men with lymph node or distant metastases at diagnosis will be excluded from the analysis.

Each of the hypothesised prognostic factors for prostate cancer mortality identified a priori (see table 1) will be recorded as an 'exposure' if it is entered into the patient record at the study baseline (index date), or recorded before the diagnosis of prostate cancer is entered. Continuous variables, such as height, weight and the biochemical markers, will be measured according to the most recent result prior to the coding of a diagnosis of prostate cancer within the study time period. Genetic factors, lifestyle exposures, medications and comorbidities will be considered in a binary manner in relation to their presence or absence at the index date. Missing data will be controlled for using multiple imputation methods.[48]

To achieve 95% power and detect a difference in HRs of 0.5 in prostate cancer mortality for a binary risk factor using an alpha of 0.05, a sample of at least 8762 men with prostate cancer would be required, assuming a 1% annual mortality rate over a median 10-year follow-up.

The primary outcome measure will be prostate cancer mortality. Participants will be presumed to be alive at the end of the follow-up period if they have not been reported as deceased according to the ONS mortality

| Table 1 | Prognostic factors to be assessed | |
|---|---|---|
| **Category** | **Prognostic factor(s)** | **Definition/unit** |
| Basic demographics | Age and date of birth | Years |
| | Post code | GP practice address |
| | Ethnicity | ONS ethnicity categories |
| Physiological | Height | Centimetres (cm) |
| | Weight | Kilograms (kg) |
| | Waist circumference | Centimetres (cm) |
| | Waist:hip ratio | |
| Genetic | Family history of prostate cancer | Recorded diagnosis in first-degree or second-degree relative |
| Biochemical | Triglycerides, HDL cholesterol, LDL cholesterol and VLDL cholesterol | mmol/L |
| | PSA | µ g/L |
| | HbA1c | mmol/L |
| | CRP | mg/L |
| | Ferritin | µg/L |
| | Haemoglobin | g/L |
| | Albumin | g/L |
| | Serum glucose and plasma glucose | mmol/L |
| | Lead | µg/L |
| Lifestyle | Smoking history | Smoking tobacco prior to or at index date |
| | Relationship status | Patient identifies as being in a relationship |
| | Alcohol intake | Units per week |
| Medications | Simvastatin, atorvastatin, metformin, aspirin, atenolol, bisoprolol, sotalol, labetalol, carvedilol, nebivolol, metoprolol, propranolol, finasteride, dutasteride, cholecalciferol, ergocalciferol and alfacalcidol | Prescribed within the 12 months prior to index date |
| Comorbidities | Type 2 diabetes mellitus, ischaemic heart disease, stroke, peripheral vascular disease, benign prostatic hypertrophy and COPD | Diagnosed prior to index date |

COPD, chronic obstructive pulmonary disease; CRP, C reactive protein; GP, General Practitioners; HbA1c, haemoglobin A1c; HDL, high-density lipoprotein; LDL, low-density lipoprotein; ONS, Office for National Statistics; PSA, Prostate Specific Antigen; VLDL, very-low-density lipoprotein.

data. Secondary outcome measures of prostate cancer progression will include all-cause mortality, change from localised to metastatic disease, and commencing antiandrogen therapy or chemotherapy. We will use whether the treatment recorded in the registry is stated to be localised (ie, one tumour treated) or systemic (ie, more than one tumour treated) to help distinguish between early and advanced disease.

Descriptive statistics will be used to summarise the basic demographic details of the men. The prevalence of the preselected putative prognostic factors will be calculated and presented. Cox proportional hazards regression will be used to estimate the crude and mutually adjusted HRs (with 95% CI) for prostate cancer and all-cause mortality according to the prognostic factors. Related prognostic factors, such as smoking and chronic obstructive pulmonary disease, will also be grouped to account for potential intermediate variables. This analysis will be repeated with stratification by stage at diagnosis. In order to allow for flexibility in the shape of the cumulative hazard function, we will use flexible parametric survival models[49] for prognostic modelling. These models incorporate cubic spline terms in the log cumulative hazard function and are based on weibull, loglogistic or lognormal distributions of survival time. We will check for non-linearities in the effects of continuous predictors using fractional polynomials[50] and also test for time-varying effects of prognostic factors. We will determine mortality risk in groups defined by important prognostic factors. To asses competing risks, we will use cause-specific survival analysis to estimate at 1, 2, 5 and 10 years post prostate cancer diagnosis the contribution of

prostate cancer mortality to overall mortality in those who have died by prognostic factor combinations.[51]

## ETHICS AND DISSEMINATION

This study protocol has approval from the Independent Scientific Advisory Committee for the UK Medicines and Healthcare products Regulatory Agency Database Research (protocol 17_041).

The findings of this study will be submitted as a manuscript to peer-reviewed journal to aid dissemination to clinicians and other researchers in the field. It will also be presented and discussed at local CPRD working groups to inform other researchers' methods using the CPRD database. Subsequent studies of the prediction tool, based on this piece of research, will involve clinicians at every stage to ensure the final tool is acceptable for use in clinical practice.

## CONCLUSIONS

This study will lay the foundation for the development of a clinically useful risk prediction tool. Clinicians will be able to use the tool, inputting routine primary care data, to improve shared decision-making about an individual's prognosis and, if validated and shown effective in trials, inform their practice when deciding with patients whether to undergo radical surgery or radiotherapy or be followed up conservatively using active monitoring.

Patients will also benefit from this work in other ways. They will be able to receive more information from GPs and NHS specialists about the risk of progression of their prostate cancer, and they will be able to decide within a shared decision-making framework with their doctors about the potential benefits and harms of undergoing radical treatment or active monitoring.

**Contributors** SWDM: performed the literature review that informed this protocol; drafted the introduction, methods and dissemination sections. RMM and MTM: drafted the statistical analysis outline. All authors: reviewed the draft manuscript, commented and amended it, and approved the final submission.

**Funding** This project is in part funded by the NIHR CLAHRC West. SWDM is the recipient of an NIHR academic clinical fellowship. RMM is supported by a Cancer Research UK Programme Grant, the Integrative Cancer Epidemiology Programme (C18281/A19169).

**Disclaimer** The views expressed are those of the authors and not necessarily those of the NHS, the NIHR or the UK Department of Health.

**Competing interests** RMM has received grants from Cancer Research UK in the previous 3 years.

**Patient consent** Not required.

**Provenance and peer review** Not commissioned; externally peer reviewed.

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
