## [Reviewer comments · BMJ Open]

ARTICLE DETAILS

TITLE (PROVISIONAL)	Predicting prostate cancer progression: protocol for a retrospective cohort study to identify prognostic factors for prostate cancer outcomes using routine primary care data.
AUTHORS	Merriel, Samuel; Martin, Richard; May, Margaret

VERSION 1 – REVIEW

REVIEWER	Michael Goodman Emory University, USA
REVIEW RETURNED	03-Oct-2017

GENERAL COMMENTS	This protocol intends to address an important area of prostate cancer research – the roles various patient and health-related factors as determinants of prognosis. The particular strengths of this protocol include reliance on well-established data sources with large numbers of cases, and ability to collect information systematically, without the need for patient opt-in. The overall strategy and design of the study are reasonable; however, there are some questions about implementation. These are listed below: 1. It is not entirely clear how some of the factors will be ascertained. I am not familiar with CPRD structure, but I would assume some data elements have corresponding fields and some need to be obtained from the notes. For example, what is the source of information for family history? Would it be possible to extend Table 1 by adding another column that lists data sources and definitions?2. It is a bit surprising that Table 1 does not list any of the prostate-cancer prognostic factors such as stage and grade. I understand that those will be obtained from the cancer registry data. Would stage data reflect clinical or pathology assessment or both? Also, would Gleason score data pertain to biopsies or surgeries or both?3. What type of information is available on the details of prostate cancer treatment? Are the data on type of surgery and radiation? What about hormonal therapy?4. The Methods do not mention collection of PSA data. PSA monitoring is a critical element of post-treatment follow up. Are these data available?5. The outcomes of interest are prostate cancer mortality and disease progression. The latter is defined as change from localized to metastatic disease, and initiation of chemo- or androgen deprivation therapy. Would the authors also consider PSA recurrence, which is a more sensitive measure of prostate cancer progression?
--

	6. An important issue in survival analysis is validation of vital status for participants presumed to be alive. In the US this is addressed via active follow up to document date of last contact. How is this issue handled in the protocol? In other words what would be the last date of follow up for subjects presumed to be alive? 7. Following protocol implementation, the resulting data will likely permit a multitude of analyses addressing a variety of research questions. Each of these research questions may require a different statistical modeling approach and different strategies for selecting and handling covariates or effect modifiers. Would the authors consider changing the emphasis in the Methods from statistical analyses to data collection and validation procedures? It is the quality and quantity of the data that are of greater interest to the readers.
--	---

REVIEWER	Bettina F. Drake Washington University School of Medicine USA
REVIEW RETURNED	23-Oct-2017

GENERAL COMMENTS	Methods: What is the average length of follow-up? How long were men followed for death post diagnosis (What is the timeframe death data was collected)? Describe how/why the specific medications and comorbidities were chosen as covariates to be collected. Is anyone considered lost-to-follow-up?
---

VERSION 1 – AUTHOR RESPONSE

Reviewer: 1

Reviewer Name: Michael Goodman

Institution and Country: Emory University, USA

Competing Interests: I have no competing interests

This protocol intends to address an important area of prostate cancer research – the roles various patient and health-related factors as determinants of prognosis. The particular strengths of this protocol include reliance on well-established data sources with large numbers of cases, and ability to collect information systematically, without the need for patient opt-in.

Thank you for your positive comments regarding this manuscript

The overall strategy and design of the study are reasonable; however, there are some questions about implementation. These are listed below:

1. It is not entirely clear how some of the factors will be ascertained. I am not familiar with CPRD structure, but I would assume some data elements have corresponding fields and some need to be obtained from the notes. For example, what is the source of information for family history? Would it be possible to extend Table 1 by adding another column that lists data sources and definitions?

Response: The CPRD dataset comprises data extracted directly from the electronic healthcare records of patients registered with participating family medicine (GP) clinics, based on Read codes

embedded within the software. Binary items, such as the presence of a co-morbidity or the family history of a particular disease, are coded as present if entered into the record by the patient's GP. Continuous variables, such as weight or HbA1c, are entered directly into the health record or imported from the local pathology service.

The data sources comprise the electronic health record(CPRD), with linked datasets from the Office for National Statistics (ONS) mortality data, and national cancer registry data.

Table 1 has been amended to define the exposure variables.

2. It is a bit surprising that Table 1 does not list any of the prostate-cancer prognostic factors such as stage and grade. I understand that those will be obtained from the cancer registry data. Would stage data reflect clinical or pathology assessment or both? Also, would Gleason score data pertain to biopsies or surgeries or both?

Response: Both clinical and pathological TNM staging and Gleason grading (from biopsies) will be obtained from the cancer registry. In the event of incomplete data, the cancer registry provides a 'best' TNM stage and Gleason grade based on the available data. The Methods section has been amended to give further information about data obtained.

3. What type of information is available on the details of prostate cancer treatment? Are the data on type of surgery and radiation? What about hormonal therapy?

Response: Prostate cancer treatment data is being provided by the linked national prostate cancer registry dataset. We will obtain the following data;

Surgery: date of procedure, type of surgery performed, number of tumours removed

Radiotherapy: date of treatment, type of radiotherapy

Chemotherapy: date of treatment, type of chemotherapy

Prescription of hormonal therapy is usually recorded in the primary care record, and as such should be available from the CPRD database.

4. The Methods do not mention collection of PSA data. PSA monitoring is a critical element of post-treatment follow up. Are these data available?

5. The outcomes of interest are prostate cancer mortality and disease progression. The latter is defined as change from localized to metastatic disease, and initiation of chemo- or androgen deprivation therapy. Would the authors also consider PSA recurrence, which is a more sensitive measure of prostate cancer progression?

(Response to 4. and 5.) This study is focused on identifying prognostic factors for prostate cancer progression present at the time of diagnosis. As you say, PSA is used for post-treatment monitoring and is a sensitive measure, but unfortunately this data is not routinely recorded in primary care in the UK and would not be available through CPRD.

As such we were unable to include PSA as a marker of progression in this particular study. The secondary outcome measures relating to progression of prostate cancer that we are measuring have been obtained through linked cancer registry data.

6. An important issue in survival analysis is validation of vital status for participants presumed to be alive. In the US this is addressed via active follow up to document date of last contact. How is this

issue handled in the protocol? In other words, what would be the last date of follow up for subjects presumed to be alive?

Response: Participants will be assumed to be alive if they have not been reported as deceased according to the ONS mortality data, which is known to have strong coverage of death reporting in the UK through the General Register Office.

7. Following protocol implementation, the resulting data will likely permit a multitude of analyses addressing a variety of research questions. Each of these research questions may require a different statistical modelling approach and different strategies for selecting and handling covariates or effect modifiers. Would the authors consider changing the emphasis in the Methods from statistical analyses to data collection and validation procedures? It is the quality and quantity of the data that are of greater interest to the readers. Thank you for your thoughts and suggestions on the focus of the article.

Response: This study is specifically trying to identify prognostic factors associated with prostate cancer progression. Further analyses on developing and validating a prognostic model based on the identified prognostic factors is planned in the future.

Reviewer: 2

Reviewer Name: Bettina F. Drake

Institution and Country: Washington University School of Medicine, USA

Competing Interests: None declared

Thank you for taking time to review our manuscript.

Methods: What is the average length of follow-up? How long were men followed for death post diagnosis (What is the timeframe death data was collected)?

Response: Average length of follow-up is currently being calculated, as we have only recently received our dataset from CPRD.

Death data was collected from 01/01/1998 – 31/12/2016.

Comment: Describe how/why the specific medications and comorbidities were chosen as covariates to be collected.

Response: The authors performed a literature review to determine what prognostic factors have been shown to be associated with prostate cancer mortality (our endpoint of prostate cancer progression) to date, and then determined which of these prognostic factors could be measured within CPRD.

Comment: Is anyone considered lost-to-follow-up?

Response: Loss to follow-up is a potential issue in the running of a cohort study, but the authors believe this will be limited by using the CPRD dataset. In the UK, approximately 98% of citizens are registered with a GP practice, giving the dataset a very high level of coverage. The major reasons that a patient would be lost to follow-up in this study would be if they died, changed GPs, or did not attend their GP practice for years during the study period.